# Systematic and quantitative view of the antiviral arsenal of prokaryotes

Florian Tesson[1,2], Alexandre Hervé[3], Ernest Mordret[2], Marie Touchon [4], Camille d'Humières[1], Jean Cury [2,5✉] & Aude Bernheim [1,2✉]

Bacteria and archaea have developed multiple antiviral mechanisms, and genomic evidence indicates that several of these antiviral systems co-occur in the same strain. Here, we introduce DefenseFinder, a tool that automatically detects known antiviral systems in prokaryotic genomes. We use DefenseFinder to analyse 21000 fully sequenced prokaryotic genomes, and find that antiviral strategies vary drastically between phyla, species and strains. Variations in composition of antiviral systems correlate with genome size, viral threat, and lifestyle traits. DefenseFinder will facilitate large-scale genomic analysis of antiviral defense systems and the study of host-virus interactions in prokaryotes.

[1] Université de Paris, IAME, UMR 1137, INSERM, Paris, France. [2] SEED, U1284, INSERM, Université de Paris, Paris, France. [3] Independent Researcher, Paris, France. [4] Institut Pasteur, Université de Paris, CNRS, UMR3525, Microbial Evolutionary Genomics, Paris 75015, France. [5] Université Paris-Saclay, CNRS, INRIA, Laboratoire Interdisciplinaire des Sciences du Numérique, UMR, 9015 Orsay, France. ✉email: jean.cury@normalesup.org; aude.bernheim@inserm.fr

Prokaryotes have evolved multiple lines of defense against their viruses. Up to 2018, only a few prokaryotic antiviral systems were described, including CRISPR-Cas systems, Restriction-Modification (RM) and Abortive infection (Abi). In 2018, a landmark study marked the beginning of a new era of discovery by revealing the existence of ten novel antiviral defense systems[1]. Since then, dozens of novels systems have been unearthed. A majority of these systems were uncovered through the "defense islands" method, using a guilt by association approach[1–8]. Others were discovered individually[9–11] or by looking into hotspots encoded in mobile genetic elements[12–14]. It is thus now recognized that prokaryotic immunity is much more complex than previously perceived with evidence for intracellular signaling regulating defense[5,15], chemical defense[7,16], nucleotide depletion[8], RNA mutations[4], guardian systems[6] and the discovery of many prokaryotic defense systems which mechanisms are still unknown[1,4,12].

The discovery of a stockpile of novel antiviral systems questions our view of how prokaryotes defend themselves against viruses. While many families of systems have been investigated mechanistically, much remains to be uncovered about the antiviral arsenal at the level of a strain, a species or all prokaryotes. Describing which systems are present in a genome will be essential for understanding phage bacteria interactions, in a natural context.

Establishing a holistic, genome centric view of the whole antiviral arsenal of prokaryotes is currently challenging. Most studies tackling the distribution of defense systems focused on one or a few families of systems. They provide important numbers regarding their abundance in microbial genomes. For example, in a dataset of 38,167 genomes, 4894 encoded CBASS (13%) and 4446 retrons (11%)[6,17]. Systems described by Doron and colleagues[1] were found at frequencies ranging from 1.8% (Kiwa) to 8.5% (Gabija)[1] while RM and CRISPR-Cas are encoded by 74.2%[18] and 39%[19] of genomes respectively. However, for many systems, the frequency remains to be studied.

One of the reasons for such lack of knowledge can be attributed to the absence of a tool dedicated to the genomic detection of known prokaryotic antiviral systems. Programs exist for the detection of specific systems such as CRISPR-Cas[20–23], as well as databases of anti-phage systems such as REBASE for RM[24] and databases of defense genes such as PADS[25]. However, a single defense gene may not be enough to characterize a functional antiviral mechanism, therefore it appears more relevant to search for complete systems. Very recently, an additional tool PADLOC[26] was published, dedicated to the detection of defense systems and allowing detection of systems uncovered by Doron and colleagues[1] as well as CBASS systems[5,17] (in total 12 systems). The lack of tool able to find all known antiviral systems is explained partly by the timeline of the discoveries of the novel systems (mostly since 2018), by their large number (more than 50) and the complex biology of defense systems.

In this study, we developed a tool, DefenseFinder, to detect known prokaryotic antiviral systems from a genomic sequence. We used this tool, to detect all known antiviral systems in a database of more than 21,000 complete microbial genomes, describe and analyze their distribution at different phylogenetic scales (from the genome to microbial species and phyla) to provide a systematic and quantitative view of the antiviral arsenal of prokaryotes.

## Results

**DefenseFinder, a tool to automatically detect known prokaryotic antiviral systems.** We set out to build DefenseFinder, a tool to detect all known prokaryotic antiviral defense systems in a

given genomic sequence. To do so, we used MacSyFinder[27], a program dedicated to the detection of macromolecular systems. MacSyFinder functions using one model per system. Each model operates in two steps (Fig. 1): first the detection of all the proteins involved in a macromolecular system through homology search using HMM profiles; second, a set of decision rules is applied to keep only the HMM hits that satisfy the genetic architecture of the system of interest. This two-steps approach is perfectly adapted for the detection of antiviral systems, which can exist under different genetic architectures. In fact, it has already been used for the detection of CRISPR-Cas systems[22,27]. We thus built or re-used the HMM profiles of proteins involved in defense systems, and defined specific decision rules for each known antiviral system (Fig. 1a).

The building of DefenseFinder required an exhaustive literature search of known antiviral mechanisms. In this version of DefenseFinder, we included all described systems discovered before November 2021 (see Methods). We excluded some systems, such as Argonautes and Toxin-Antitoxin (unless a specific family role had been demonstrated such as DarTG[28]), as it is unknown whether all members of such families are involved in antiviral defense. In total, DefenseFinder detects 60 antiviral families (Supplementary Data 1 and 2). When available, we also included the types and subtypes of different systems (e.g. CBASS type I, Retron IV, CRISPR-Cas I-E), leading to a total of 151 (sub) types of systems (Fig. 1a, Supplementary Data 1 and 2).

For each of the 151 systems, we defined a model e.g. a set of customized rules and associated protein profiles (see Methods). Briefly, we either used existing pfams/COGs or built custom HMM profiles resulting in a database of 845 profiles (Supplementary Data 2). The decision rules are typically defined by a list of mandatory, accessory, or forbidden proteins necessary for the detection of a given system (Fig. 1c). For each of the proteins, several homologs can be interchanged. For example, for CBASS type I, two mandatory proteins are necessary (the cyclase and the effector) and a third one is accessory. Each of these proteins has several possible homologs (Effector can be 1TM, 2TM, phospholipase….).

Once a first model is defined, when possible, we evaluate it, against existing datasets[1,19,24,29] (See Methods, Supplementary Figs. 1–4). For example, for the 10 systems described by Doron and colleagues, where a detailed detection was provided for each system, we downloaded the same database of genomes and used our models to search for these systems. We could then compare our detection and evaluate which systems were for example missing. The model (either profiles or decision rules) could then be adapted (see Methods sections). Using similar types of approaches, models were improved, and we could report a sensitivity for such systems, ranging between 97.4% (DISARM) and 99.4% (Septu) and a high specificity, ranging between 96.7% (Septu) and 99.97% (Thoeris) (Supplementary Figs. 1–3, Supplementary Data 3). Detection results were then compared with another program dedicated to the detection of anti-phage systems, PADLOC[26] on the 12 anti-phage systems that PADLOC detects (Supplementary Fig. 4). Both programs were in accordance in a vast majority of cases, ranging from 76% for Septu to 97,7% for Wadjet (Supplementary Fig. 4). Details about validation process for other systems are found in the methods section. All models are under a CC-BY-NC license and available online (https://github.com/mdmparis/defense-finder-models).

The final step was to use such rules and profiles to build a user-friendly tool to detect antiviral systems in prokaryotic genomes (Fig. 1d). We provide a command line interface (easily installable through a python package mdmparis-defense-finder) as well as a webservice (https://defense-finder.mdmparis-lab.com/). Both take as an input a protein multifasta file (either one or several genomes

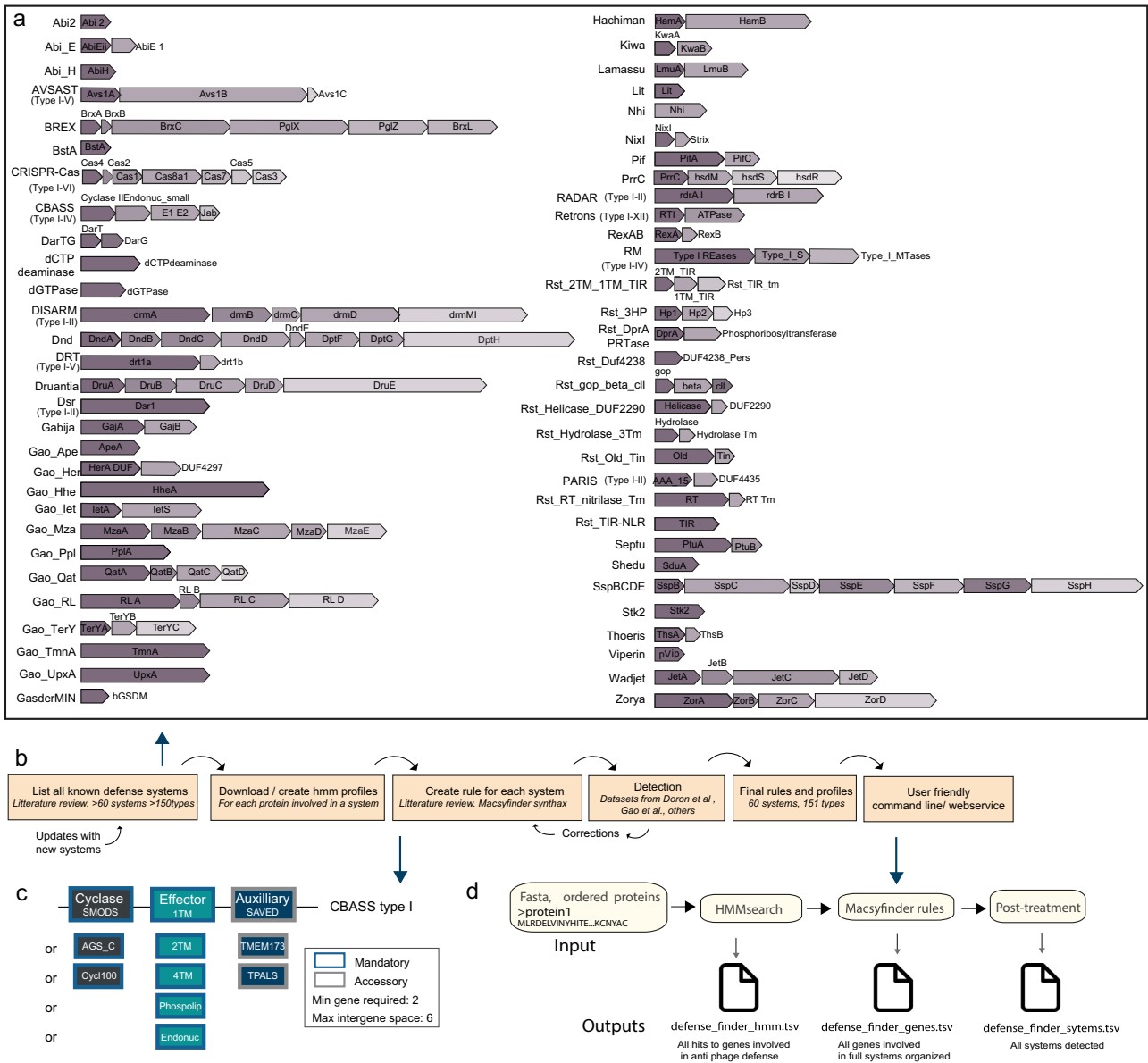

**Fig. 1 DefenseFinder, a tool to detect all known prokaryotic antiviral systems. a** List of systems included in DefenseFinder. Systems are ordered alphabetically. For systems with several types, each system is represented by one type and other types are indicated in parentheses (Full list, Supplementary Data Table 1). **b** Workflow for the creation of DefenseFinder. **c** Example of a DefenseFinder rule (in the MacsyFinder syntax) for the detection of system CBASS type I. Cyclase and effector proteins are mandatory while the sensing protein is only accessory. This means the system allows for it to be missing in a detected CBASS type I system. Different profiles are recognized for a protein. ex cyclase (SMODS, AGC_C, Cycl100). **d** DefenseFinder function layout. DefenseFinder takes an ordered multifasta protein file. A search for specific HMM profiles is conducted, the MacSyfinder rules specific for antiviral systems are applied on the search results, generating three results files.

at the same time) or a nucleic fasta file for the webservice and generate two types of detection files: a list of detected systems and a list of proteins involved in detected systems. The online service also offers data visualization of the results. The current settings of DefenseFinder are optimized for a conservative detection, as only full systems are present. This can lead to an underdetection of some proteins involved in antiviral defense. Typically, in defense islands, full systems along single proteins (otherwise involved in an antiviral defense system) are present. To overcome this, we also propose as an optional output, a list of all the hits to known antiviral proteins, which can allow a more exhaustive vision of the potential proteins involved in antiviral defense. The architecture of DefenseFinder is designed for easy and frequent updates of the models, a necessity in the fast-evolving field of antiviral defense.

The webservice will use the most up-to-date rules, and the command line interface can get the most up-to-date rules by calling the option "–update". The update on the command line is distinct of the update of DefenseFinder allowing to have update of the rules and profile more frequently. In summary, we created DefenseFinder, a program that enables the detection of all known antiviral systems in prokaryotic genomes.

**The antiviral arsenal of prokaryotes is highly variable.** In order to provide a genome-centric view of the antiviral arsenal of prokaryotes, we applied DefenseFinder to a database of 21,738 fully sequenced prokaryotic genomes including 21,364 bacterial and 374 archaeal genomes (Supplementary Data 4). We identified

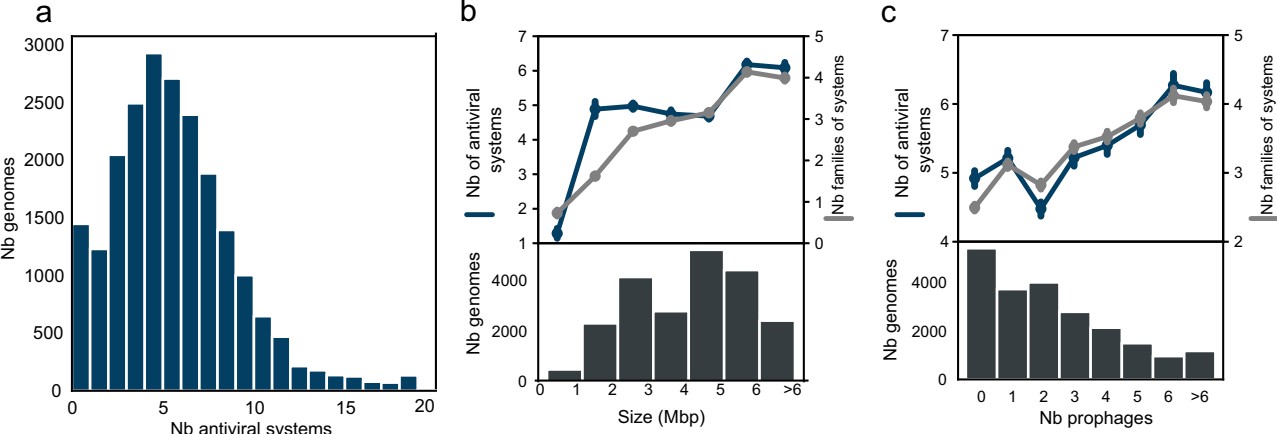

**Fig. 2 Distribution of antiviral systems in prokaryotic genomes. a** Distribution of the total number of antiviral systems per genome. The x-axis was cut at 20 for data visualization purposes. Max number is 57 (*Desulfonema limicola*). **b** Average number of antiviral systems (blue) and number of families of antiviral systems (gray) are correlated with genome size (two-sided Spearman, $\rho = 0.25$ and $\rho = 0.44$, both p-values < 0.0001 on all sizes; $\rho = 0.27$ and $\rho = 0.30$, p-value < 0.0001 for the 2 Mbp–5 Mbp interval) (CI = 95%). Bottom plot, distribution of the genome size. **c** Average number of antiviral systems (blue) and number of families of antiviral systems (gray) are correlated with number of prophages (two-sided Spearman, $\rho = 0.16$ and $\rho = 0.27$, both p-values < 0.0001) (CI = 95%). Bottom plot, distribution of the number of prophages in genomes.

113,955 different antiviral systems, comprised of 301,372 genes (Supplementary Data 5 and 6). On average prokaryotes encode five antiviral systems (5.2). The number of antiviral systems per genome varies widely from a minimum of zero (1450 genomes) to a maximum of 57 in the deltaproteobacteria *Desulfonema limicola*. (Fig. 2a, Supplementary Data 7). A large fraction of the genomes without any defense systems (74%) seems to belong to species/genus devoid of any system (Supplementary Data 9). Overall, most genomes (78%) encode more than two defense systems.

To estimate the diversity of this antiviral arsenal, we computed the number of distinct families of antiviral systems per genome (e.g., RM, CRISPR-Cas, CBASS, Supplementary Data 1). On average, prokaryotes encode three distinct families of antiviral systems (Supplementary Fig. 5). For example, *Rhodococcus opacus* B4 encodes five systems and three families (two RM, two CRISPR-Cas and a Wadjet). The number of antiviral systems in a genome correlates positively with the number of families of defense mechanisms in that genome (Supplementary Fig. 5, Spearman, $\rho = 0.79$, P-value < 0.0001) meaning that when a genome encodes a large number of antiviral systems, it is likely that these antiviral systems belong to different families of antiviral systems. There are some exceptions, with genomes encoding many antiviral systems but with "low diversity", i.e., a small number of distinct antiviral systems families. For example, *Chloroflexus aggregans* DSM 9485 encodes 17 systems but only four families of anti-phage systems. This type of genomes typically encodes a wide diversity of RM systems. Indeed, for 92% of the genomes that encode more than 10 antiviral systems belonging to four families or less ($n = 520$), RM systems make up for more than 50% of antiviral systems. Moreover, 43% of these genomes are from the species *Helicobacter pylori*, which has been described in the past as carrying many RM systems[18].

We then set out to understand the potential drivers of the number of antiviral systems in a given genome. The genome size is an important determinant for encoding accessory systems in prokaryotes. It was demonstrated that small genomes encode few CRISPR-Cas and RM[18,19]. We found this observation can be generalized to the total number of antiviral systems (Fig. 2b blue). The size effect is not linear. Very small genomes (<2 Mbp) encode few defense systems, and larger genomes encode more defense systems. However, there is no size effect observed for genomes

between 2 Mbp and 5 Mbp (Spearman $\rho = 0.07$, p-value < 0.0001). We observed a stronger positive correlation between the number of families of antiviral systems and the genome in this interval (Fig. 2b gray, Spearman $\rho = 0.30$, p-value < 0.0001).

We reasoned that the number of antiviral systems might also be influenced by the diversity of virusesa prokaryote might encounter. This can be estimated by focusing on the number of prophages encoded in a given genome. We thus focused on the interplay between antiviral systems and prophages. To do so, we detected prophages in the genomes of our database using Virsorter2[30] (see Methods). We found 51,582 prophages in 16,315 genomes, with on average two prophages per genome (Supplementary Fig. 6, Supplementary Data 10) which is in line with previous detection[31]. An intuitive hypothesis would be to expect a negative correlation between the number of antiviral systems and of prophages which would reflect the capacity of these antiviral systems to limit viral infection. However, previous studies on CRISPR-Cas and RM[18,32] demonstrated otherwise. Indeed, there was either no correlation or a positive one between the number of prophages and the antiviral system studied. We observed that the number of prophages correlates positively both with the number of systems (Spearman $\rho = 0.16$, p-value < 0.0001) and the number of families (Spearman $\rho = 0.27$, p-value < 0.0001). We controlled for the effect of the genome size on these parameters using a stepwise forward regression. Both the number of prophages and antiviral systems are still significantly correlated when accounting for genome size (p-values < 0.0001). These results suggest that the antiviral arsenal of prokaryotes is highly variable and influenced by both genomics traits, such as genome size (for small and large genomes), and life styles traits, such as the number of prophages present in the genome.

**Families of antiviral systems have a heterogeneous distribution**. We then decided to inspect the distribution of antiviral systems individually (Fig. 3a). RM systems are by far the most abundant as they are present in 83% of prokaryotic genomes, followed by CRISPR-Cas (39%). Apart from these systems, the frequency of the most abundant system drops below 20% (Gabija, Wadjet, Retrons, CBASS, AbiEII, Abi2) ranging from 10 to 17% genomes encoding such systems. The frequencies we find are

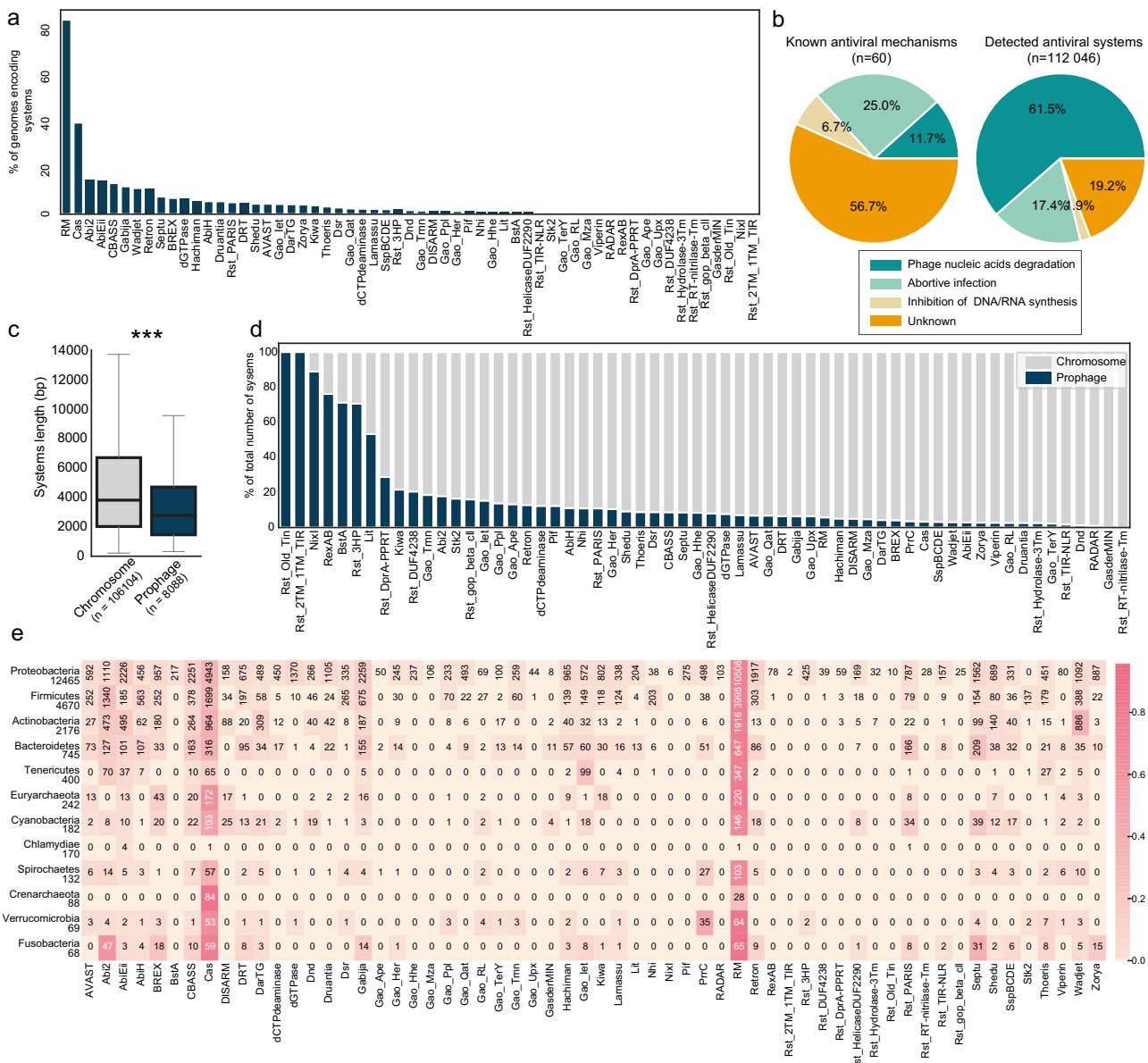

**Fig. 3 Families of antiviral systems have a heterogenous distribution. a** Frequency of systems in genomes. **b** Classification of antiviral mechanisms and detected operons according to their molecular functioning. **c** System length according to genomic location (***, two sided Wilcoxon test $p$-value $= 6.5e{-}273$)). The box extends from the first quartile (Q1) to the third quartile (Q3) of the data, with a line at the median. The whiskers extend from the box by 1.5× the inter-quartile range (IQR). $n$ chromosome $= 106,104$, $n$ prophage $= 8088$. Plasmids were excluded from this analysis. A system was deemed to be part of a prophage when the first and last protein of the system was inside the prophage boundary. The size of a system was computed as the difference between the end of the last protein and the beginning of first protein. **d** Frequency of genomic location for each system. **e** Number of systems per prokaryotic phylum. Only phyla with more than 50 genomes are represented. The number of genomes in the dataset is represented under each phylum's name. The heatmap represents the frequency of each system in a given phylum (per line), color legend on the right. Absolute numbers of genomes encoding a given system is indicated in each cell.

consistent with what has been described previously[1,6,17–19]. Among the 60 families detected in this study, 21 are present in less than 1% of the genomes. Although these systems are very rare, they are still present in diverse species and genera. Only four systems were found in less than 10 different genera (Supplementary Data 8). Such numbers suggest that the diversity of antiviral systems is enormous, and many rare systems exist. As RM are the most abundant systems, we checked whether genomes without RM had specific antiviral mechanisms (Supplementary Fig. 7). The most abundant systems in these genomes correspond to the most abundant systems in all prokaryotic genomes suggesting that no specific system replaces RM ones.

In order to understand the main mechanisms used by prokaryotes to defend themselves against viruses, we classified antiviral mechanisms in three categories based on what is known is the literature (virus nucleic acid degradation, Abortive infection, inhibition of DNA/RNA synthesis, unknown, Supplementary Data 1). For example, CRISPR-Cas and RM function through virus nucleic acid degradation, while CBASS and Retrons function through abortive infection and Viperins and dCTPdeaminase through inhibition of DNA/RNA synthesis. Overall, we found that while 58% of the described combinations of genes in the literature to compose "antiviral system" function in an unknown manner, 63.2% of the detected systems in genomes

function through nucleic acid degradation (Fig. 3b). Thus, while still many more mechanisms might be discovered, nucleic acid degradation appears as the major mode of antiviral defense in prokaryotes.

Following, the observation that systems' distribution is contrasted, we set out to understand drivers of such heterogeneity. Several systems recently described were discovered in prophages or their parasites[10,12,14,33,34]. These observations could suggest that systems encoded on prophages differ than those encoded in the chromosome. We thus evaluated the antiviral systems based on their genomic location either prophage-encoded ($n = 8088$) or chromosomal ($n = 106,070$). It was recently proposed that systems encoded in P2 and P4 (prophages of *E. coli*) could be shorter than other common systems such as CRISPR-Cas[12]. We checked whether this observation could be generalized. Systems located within prophages are shorter (Fig. 3c, median prophage-encoded systems = 2569 bp, median chromosomal systems = 3596 bp, Wilcoxon rank sum test, *p*-value < 0.0001). This is compatible with the size constraints exerted on such elements and suggest that different systems are encoded on the chromosome and prophages. To check this hypothesis, we computed for each system the frequency of the location (either prophage-encoded or chromosomal) (Fig. 3d). For example, out of the 1019 Kiwa, 799 are encoded on the chromosome, and 220 in prophages (21%). We found that some systems such as NixI, Rst_3HP, BstA, Rst_Old_Tin, Rst_2M_1M_TIR and RexAB are encoded in majority on prophages. Incidentally, these were discovered in prophages and phages[12,14,35]. Some systems such as Dnd or RADAR are almost never found on prophages. The large size of certain systems (Dnd, BREX, DISARM) could explain their absence on prophage. While our results are restricted to prophages and not their satellites, they suggest that some systems might be prophage specific.

We then evaluated if phylogeny affects the distribution of antiviral systems and thus examined the distribution of individual antiviral systems per phylum (Fig. 3e). First, a striking feature is the quasi absence of any anti-phage systems in Chlamydiae as observed in previous studies[17–19,29]. This could be explained by the intracellular lifestyle of such bacteria. More generally, genera without any or very few systems correspond to genera of bacteria that are obligate intracellular or endosymbiont (Supplementary Data 9). Alternatively, these bacteria could encode anti-phage systems that are currently unknown. The most widespread systems (RM, CRISPR-Cas) are present and quite abundant in most phyla (RM > 78% except Chlamydiae and Crenarchaeota; CRISPR-Cas > 36% except for Chlamydiae and Tenericutes). Some systems, while less abundant, are widespread across all phyla (such as Abi2, Retrons, CBASS, Hachiman, AVAST…) whereas some other systems are enriched in specific phyla. Typically, many systems such as BstA, RexAB, Gao_Hhe or Pif are only present in Proteobacteria. This might be explained by a limited detection capacity by our models or by a bias of the database towards proteobacterial genomes ($n = 12,465$), which represents 57% of the database; it could indicate the existence of more phylogenetically restricted systems, as most of these systems were discovered in prophages of Proteobacteria. Finally, some systems seem to be particularly enriched in specific phyla such as BREX in Fusobacteria (Frequency of 26.5% compared to 7% for all prokaryotes) or Wadjet in Actinobacteria (41% compared to 11% for all prokaryotes). Overall, our results demonstrate that the distribution of antiviral systems is heterogeneous and influenced by genomic location and phylogeny.

**The antiviral arsenal of bacteria is species specific.** Following our observation that some antiviral systems are enriched in specific phyla, we decided to focus on the link between antiviral defense and phylogeny. To do so, we examined the differences between the antiviral arsenal of diverse species. We selected all the species with more than 100 genomes in our dataset (nb = 21, see Methods) and established a quantitative comparison between their antiviral arsenal (Fig. 4, Supplementary Figs. 8 and 9). Both total numbers of systems and number of different families varies widely between species. For instance, we found no anti-phage system in the 558 genomes of *Bordetella pertussis*, but 18 anti-phage systems on average in *Helicobacter pylori* strains. This is in line with previous reports of *H. pylori* encoding many RM systems[18]. We found species encoding few systems such as *Bacillus subtilis*, *Staphylococcus aureus*, *Streptococcus pyogenes* (average number of systems < 3) and species encoding many anti phage systems such as *Escherichia coli*, *Pseudomonas aeruginosa*, or *Neisseria meningitidis* (nb > 6). Number of systems correlates with diversity of systems within a species, except for three species (*H. pylori*, *N. meningitidis*, *C. jejuni*) with many systems (nb > 6) but only a few families (nb < 3). Overall, these results suggest that species have very diverse types of defense arsenal which could be grouped in three categories: few systems, many diverse systems, many similar systems (Supplementary Fig. 9a).

To better understand the different types of antiviral arsenals, we decided to characterize the anti-phage systems distribution in 15 of these species (See Methods, Fig. 3, Supplementary Figs. 8 and 9). For each species, we computed (1) the distribution of the total number of systems to evaluate if this was a conserved trait across a species, (2) The frequency of the 20 most common antiviral systems in prokaryotes in this species (3) A phylogenetic tree based on the core genome of each species where the 10 most common anti-phage systems are mapped. These types of representation allow for a comparison between different species. We found very different trends for different species.

An example of the category "many diverse systems" is *P. aeruginosa* (Fig. 4b). The number of anti-phage systems varies greatly from one strain to another from one to seventeen. Similar to the global distribution, the most common systems are RM and CRISPR-Cas. However, some rarer systems such as CBASS and Gabija are present in more than 40% of the strains while some common systems such as dGTPase are absent from these genomes. Our phylogenetic analysis reveals a patchy distribution of anti-phages systems even in closely related strains, suggesting high rate of horizontal gene transfer. For example, the two closely related strains *P. aeruginosa* Ocean-1155 (GCF_002237405.1) and *P. aeruginosa* C79 (GCF_007833895.1) have a very different antiviral arsenal (respectively 7 systems with 3 RM, 1 Abi2, 1 AbiEii, 1 CBASS, 1 Gabija vs 12 systems including 5 RM, PrrC, 1 Abi2, 1 BREX, Gabija, CBASS, Shedu, and Gao_Iet). This is in line with previous observations that *P. aeruginosa* closely related strains could have diverse antiviral arsenal[36], but demonstrates that this trend concerns the entire species not a specific part of it. To control that our results were not due to biases in the number or diversity of genomes, we analyzed the correlation at the species level between the diversity in the antiviral arsenal and the phylogenetic distance (Supplementary Fig. 9b). To do so, we computed the Bray-Curtis distance of all pairs of antiviral arsenals as well as the phylogenetic distance between all strains (Supplementary Fig. 9). We found no correlation between these values for *P. aeruginosa*. We found a similar trend for other species such as *E. faecium* and *A. baumanii* (Supplementary Fig. 9b).

The study of the antiviral arsenal of *N. meningitidis* uncovered very different conclusions. The number of anti-phage systems is almost constant with a very narrow distribution centered around 8 systems. We found only four families of anti-phage systems, RM, CRISPR-Cas, Abi2 and PARIS. Contrary to *P. aeruginosa*, the

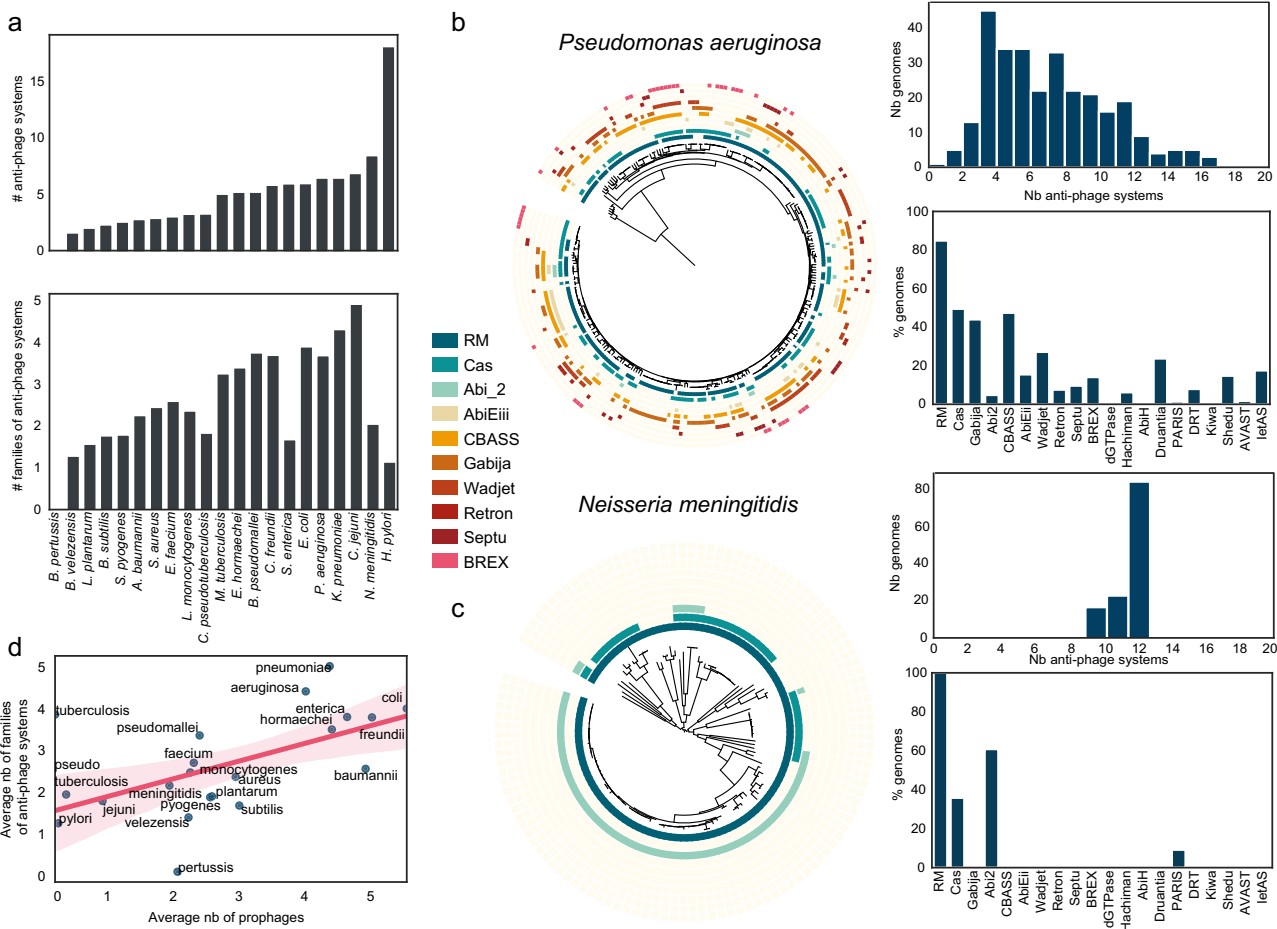

**Fig. 4 The antiviral arsenal is species specific. a** Number of systems and number of families of systems per species. Species with more than a hundred genomes were selected for this analysis. **b, c** Antiviral arsenal of two different species: *Pseudomonas aeruginosa* and *Neisseria meningitidis*. Each panel shows the distribution of the total number of systems in the species (top panel), the frequency of the 20 most common antiviral systems in prokaryotes in this species (bottom panel) and a phylogenetic tree of the species with the presence/absence of the 10 most common antiviral systems in prokaryotes. **d** Correlation at the species level between the number of prophages and the families of antiviral systems (linear regression, two sided Pearson $r = 0.56$, $p$-value $= 0.008$, light pink represents the confidence interval at 95%).

presence of anti-phage systems is very dependent on the phylogeny (Supplementary Fig. 7, spearman, $\rho = 0.68$, $p$-value $< 0.0001$). For example, 63 phylogenetically closely related strains have exactly 6 RM type II, 2 RM Type III, and 1 Abi2. We found a similar trend for species including *C. jejuni, S. pyogenes* and *S. aureus* (Supplementary Fig. 8b).

We then set out to understand what could drive such different strategies to fight viral infection. We observed that for all prokaryotes, genome size and number of prophages influence the number of anti-phage systems and of families. We postulated that these factors could influence the antiviral arsenal of species. However, at the species level, only the average number of families correlates with both genome size and number of prophages (Fig. 4e, Supplementary Fig. 10, spearman, $\rho = 0.7$, $\rho = 0.55$, $p$-values $= 0.0003$ and $0.01$), and not the average number of systems (Supplementary Fig. 10). This suggests that the diversification of the antiviral arsenal, not the number of antiviral systems, is impacted by these factors.

## Discussion

We provided here a framework for the quantitative analysis of the antiviral arsenals of prokaryotes by creating a tool to detect all known prokaryotic antiviral systems and using it to generate a

description of the antiviral arsenal of prokaryotes at diverse phylogenetic scales. Our analysis provides a global view of antiviral systems encoded in prokaryotic genomes, and reveals a high diversity of antiviral strategies across organisms.

DefenseFinder allows for the detection of all known prokaryotic antiviral systems. A novel tool, PADLOC was recently described and provides the description of several but not all antiviral systems. We hope that the availability of this tool in command line (for big data analysis) and through a web-service (for occasional detection) will allow the community to learn more about specific strains and bring insights on large datasets. To that end, DefenseFinder is quite adapted as it runs in less than a day on 5000 genomes on a regular laptop (using 4 CPU). A major challenge for such program is to adequately reflect the state of the literature in the field. Because the literature on this particular topic is ever changing, we took advantage of the architecture proposed by MacSyFinder to facilitate regular updates.

Many features of most antiviral systems are still not understood such as their diversity or molecular mechanisms. This lack of knowledge leads to challenges in the detection, as some might exist under unknown forms and would not be detected by our program. It is also possible that several systems will end up being regrouped in bigger families of antiviral systems. For example, Pycsar systems were recently described and are closely related to CBASS systems[36]

and are currently detected as a subtype of CBASS in Defense-Finder. All our models are freely and openly available, and we hope the community will propose novel and better models or point out to missing or inaccurate ones. Another hurdle of our program is the heavy reliance on computer inference, as we sometimes have only one or a couple systems that were experimentally validated. Similarly, for many systems, the accuracy of the detection could not be properly evaluated as no existing ground truth was available, which only experimental work can provide.

We detected and analyzed thousands antiviral systems in fully sequenced bacterial and archaeal genomes. We used the entire RefSeq complete genome database, which is known to be biased towards specific prokaryotes, notably to over-represent cultivable bacteria (1630 *Escherichia coli*, 917 *Salmonella enterica*, 770 *Klebsiella pneumoniae*, 590 *Staphylococcus aureus* and 558 *Bordetella pertussis*). Overall, species with more than 100 genomes represent 7561 genomes (34%) and species between 100 and 10 genomes represent 6906 genomes (31%). However, this study represents a comprehensive census of the antiviral arsenal of prokaryotes. While we know this census will change in its details as many antiviral systems probably remain to be discovered, we expect that the general trends that were observed in our study will not change much. Indeed, the most recent systems to be discovered are present in less than 15% of the genomes. Thus, the discovery of additional systems, might impact deeply the phage-host interactions for a specific strain or species, but the general numbers for all prokaryotes should not change drastically. Metrics we used in our study are also subjects to limitations. We evaluated the diversity of the antiviral arsenal using families of antiviral systems which we define as ensemble of systems with similar molecular mechanisms (RM, CRISPR, CBASS). However, much diversity notably in terms of molecular mechanisms exists within such families and could be inspected. Second, we used the number of prophages as an estimate of prokaryotic virus diversity. It has been shown that is some cases, species have few prophages but many virulent phages[32], this metrics does not take this aspect into account.

Despite these limitations, our data provides a quantitative description of the antiviral arsenal of prokaryotes which will serve to answer fundamental questions in phage biology. It confirms that only a few antiviral systems are very abundant. If many rare systems exist, it is highly possible that many more remains to be discovered. We showed that several anti-phage systems are enriched in prophages or in specific phyla. Understanding the causes of this enrichment might reveal specific evolutionary constraints imposed on this system. Another observation from this census is the diversity of antiviral arsenals at the species level. While specific enrichment in some antiviral systems had been shown for specific antiviral systems such as RM or CRISPR-Cas[19], our results suggest specific mechanisms are acting at the level of this antiviral arsenal.

We previously postulated the existence of a potential "Pan immune system"[37] where "the 'effective' immune system is not the one encoded by the genome of a single microorganism but rather by its pan-genome, comprising the sum of all immune systems available for a microorganism to horizontally acquire and use." Several recent studies reported diversity and dynamics of anti-phage systems in natural isolates of *Vibrio*[13,38–40] compatible with such hypothesis. Our current study suggests that this hypothesis could be relevant only for a subset of species, while other evolutionary strategies might be at play for others. Our results suggest that a high diversity of viruses could lead to a diversification of the antiviral arsenal. Other studies will be needed to shed light on such evolutionary dynamics which could have important implications for phage therapy.

Overall, our study provides both a tool and a census for the detection of prokaryotic antiviral systems paving the way to quantitative examination of several hypotheses in the field of anti-phage defense, such as the phenomenon of defense islands or co-occurrences between antiviral systems. Our quantitative framework will allow new genomic insights in this rapidly evolving field.

## Methods

**Choice of antiviral systems**. We chose to include all prokaryotic antiviral systems described with at least one experimental evidence of the antiviral function (before November 2021). As the field is fast evolving, we decided to also include systems described in preprints. Some systems such as Argonautes and Toxin-Antitoxin have not been included yet, as there is still some controversy if all members of such families are involved in anti-phage defense. Thus, only families with well-established roles such as DarTG[41] have been included. We also excluded host factors and superinfection exclusion mechanisms. In total, DefenseFinder detects 60 families of antiviral systems (Supplementary Data 1). While our aim is to be exhaustive, it is possible that some systems were missed, we call on the community to help us complete and correct the list.

**Building DefenseFinder HMM models**. The protein profiles used were either retrieved from existing databases (PFAM[42], COG[43]) or built from scratch when no adequate profiles existed (see below for details on the building of HMM profiles and Supplementary Data 2).

New protein profiles for the proteins involved in anti-phage systems were built using a homogeneous procedure. We collected a set of sequences from the protein family that were representative of the diversity of the bacterial taxonomy. Homologous proteins were aligned using MAFFT v7.475[44] (default options, mode auto) and then used to produce protein profiles with Hmmbuild (default options) from the HMMer suite v3.3[45]. To ensure a better detection we curated each profile manually by assigning a GA score (used with the hmmsearch option–cut_ga) (Supplementary Figs. 1–2). GA score defines the threshold above which a hit is considered significant. This threshold was determined manually by inspecting the distribution of the scores. All accession numbers for proteins used to build custom HMM profiles are available in Supplementary Data 2.

Protein scrapping was done using different methods depending on the available information about the system in the literature (Details in Supplementary Data 1). For systems from[1], dGTPase[8], dCTPdeaminase[8], BREX[3], part of Cyclic-oligonucleotide-based anti-phage signaling systems (CBASS)[17], all the reverse transcriptases of retrons, BstA[10], viperins[7] and DISARM[2], we used a subset (between 20 and 100 proteins) of the proteins available in the supplementary data of each publication. We then tested if the HMM allows for detection of all known occurrences of such proteins. If a lot of proteins were undetected, we added proteins reported in the supplementary materials but not detected through our HMM to the list of sequences for the alignment and subsequent HMM generation.

For AbiEii, AbiH, Abi2, Stk2, Pif, Lit, PrrC, RexAB, part of CBASS, part brxA, and brxB from BREX, PARIS (AAA15 and AAA21), we used PFAM available at (http://pfam.xfam.org/) or the sequence available on COG (https://www.ncbi.nlm.nih.gov/research/cog-project/). For part of BREX, DndABCDEFGH[46], we searched for proteins with this name available on NCBI and curated manually such list. For systems when only one sequence was provided such as Gao's systems[4], Rousset's systems[12], Dnd type SspBCDE, part of retrons, the protein sequence was BLASTed. Between 20 and 50 sequences with high coverage were selected. For retrons other than reverse transcriptase, we used the IMG genome neighborhood feature to get adjacent protein of the reverse transcriptase and repeated the BLAST process. For CAS systems, HMM protein profiles were downloaded from[22,47]. All hmm profiles used are available at https://github.com/mdmparis/defense-finder-models.

**Building DefenseFinder rules of detections**. We defined genetic organization rules based on the literature (Supplementary Data 1). MacSyfinder allows for two types of genetic components, "mandatory" and "accessory". Given the wide diversity of genetic organization of antiviral systems, initial rules were written differently for major types of systems. Typically, for small systems (<3 proteins), the number of mandatory proteins required were strict whereas for bigger system (such as Druantia), the number of proteins required did not always required all components to be present. For CAS systems, all models previously defined in CasFinder v2.0.2 include in CRISPRCasFinder[22] have been rewritten to be compatible with the new version of MacSyfinder v2.0rc4[27] and updated to take into account the most recently proposed nomenclature[47]. As a result, this new version CasFinder v3.0.0 (used in DefenseFinder) allows to detect 6 different types (I–VI) and 33 different subtypes. All DefenseFinder rules used are available at https://github.com/mdmparis/defense-finder-models.

**Validation of DefenseFinder models**. Following an initial design of rule for each system, we ran an initial detection and evaluated the results. When available, this initial detection was compared to other existing datasets (Supplementary Figs. 1–4, Supplementary Data 3).

For single gene systems, we choose specific GA cut using the distribution of score of each HMM (Supplementary Fig. 1). GA cuts were chosen in order to limit over detection for those systems.

Specificity and sensitivity were evaluated for each system from systems from Doron[1], CBASS[17] and DISARM[2] and are reported in Supplementary Fig. 3d and Supplementary Data 3. Sensitivity was defined as the percentage of systems detected by DefenseFinder among the systems detected in each of the corresponding dataset. Specificity was defined as the ratio between (1) the number of genomes where a system was detected by DefenseFinder in genomes where corresponding datasets had not detected any and (2) all the genomes where no systems had been detected. For CRISPR-Cas systems, detection was compared to CasFinder v2 from ref. [22]. For RM systems, we ran DefenseFinder on REBASE[24] (Supplementary Fig. 3e). For RM, we report a 91.9% sensibility (384231 systems detected out of 417666). Our profiles are underdetecting some distant forms of RM type IV (15% of the RM type IV on REBASE are missed) however, when such sequences were added to the HMM profiles, we observed an important drop in specificity which led us to the current trade-off.

Another type of verification was to evaluate the percentage of genomes with different systems and subsystems detected by DefenseFinder and compare it to previous detections (Supplementary Fig. 3). When no other datasets were available, the quality of the detection was estimated by checking different factors such as the number of occurrences found, size of proteins and systems. Each rule was thus refined through trial and error cycles to reach a final stable version.

**Availability**. DefenseFinder online is available at https://defense-finder.mdmparis-lab.com/.

DefenseFinder command line is available through *pip install mdmparis-defense-finder*.

DefenseFinder documentation is available at https://github.com/mdmparis/defense-finder.

DefenseFinder models are available at https://github.com/mdmparis/defense-finder-models.

**Data**. We analyzed 21,738 complete genomes retrieved in May 2021 from NCBI RefSeq representing 21,364 and 374 genomes of Bacteria and Archaea and respectively, 4374 and 260 species. All genomes were downloaded on NCBI website (https://www.ncbi.nlm.nih.gov/assembly), with the request "Bacteria OR Archaea" and the filter "Complete genome" and "Latest RefSeq". All accession numbers and phylogenetic information are available in Supplementary Data 4.

**Detection of antiviral systems**. We used DefenseFinder v1.0.2 (models v1.0., January 2021) to search for prokaryotic antiviral systems in the RefSeq database. To do so, we first formatted this database under a gembase format (see DefenseFinder documentation). We then ran DefenseFinder with the–db-type gembase. We provide the results of this detection in Supplementary Data 5–7.

**Phylogenetic analysis**. We used PanACoTA[48] version 1.2.0 to build phylogenies for 15 bacterial species (*Escherichia coli, Pseudomonas aeruginosa, Streptococcus pyogenes, Salmonella enterica, Listeria monocytogenes, Helicobacter pylori, Mycobacterium tuberculosis, Neisseria meningitidis, Staphylococcus aureus, Bacillus subtilis, Campylobacter jejuni, Klebsiella pneumoniae, Bacillus velezensis, Acinetobacter baumannii, Enterococcus faecium*). PanACoTA allows phylogenetic tree reconstructions based on the core genomes. For each of the species, we took all genomes under a nucleic acid format in NCBI (fna) and annotated them using prodigal (PanACoTA annotate options–cutn 10000–l90 400 –prodigal). We then computed the pangenome and coregenome (PanACoTA pangenome; PanACoTA corepers; with default parameters). Finally, we aligned the coregenome (PanACoTA align, default parameters) and computed a phylogenetic tree (PanACoTA tree, -b 1000). For this step PanACoTA, uses IQTree[49], (version 2.1.4) and the following option (iqtree -m GTR -bb1000 -st DNA).

**Detection of prophages**. Putative prophages were detected using VirSorter v2.2.2[30] (Supplementary Data 10). Results were filtered to exclude the least confident predictions (we kept max score > 0.8, size <200 kb) which might correspond to prophage remnants or erroneous assignments. Prophages were checked for diversity by clustering prophage proteins using MMseqs2[50] with a coverage and a minimum sequence identity of 90 %. 67% of prophage proteins are in a singleton cluster demonstrating that most phages in the dataset are diverse.

**Figures and statistical analysis**. Figures were made with matplotlib v3.3.2[51] and seaborn v0.11.0[52]. Data analysis and statistics analysis were done using pandas 1.1.3[53] and scipy 1.5.2[54]. Phylogenetic trees were plotted using ITOL[55]. Statistical significances were adjusted by Bonferroni correction.

**Reporting summary**. Further information on research design is available in the Nature Research Reporting Summary linked to this article.

## Data availability

The data that support the findings of this study are available in the Supplementary Information and Supplementary Data files. Accession codes for the proteins used to create the models are available in Supplementary Data 2. Accession codes for the genomes used for the analysis are available in Supplementary Data 4. The results of the analysis are available in Supplementary Data 5–10. Source data are provided with this paper. Due to the large size of Source Data for Fig. 9b, this Source Data is not directly available for download but available upon request to the authors.

## Code availability

The code used is available at https://github.com/mdmparis/defense-finder. All models used in the study are under a CC-BY-NC license and available online at https://github.com/mdmparis/defense-finder-models.

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

## Acknowledgements
We would like to thank Eduardo PC Rocha and members of MDM labs for fruitful discussion and review of the manuscript; Nitzan Tal for fruitful advice on data visualization; Adrien Bernheim for the logo design of DefenseFinder; Aurélien Hervé for the help on the webserver interface and Bertrand Neron and Sophie Abby for the development, maintenance and advice on MacSyFinder. This work was partly supported by the CRI Research Fellowship to Aude Bernheim from the Bettencourt Schueller Foundation, the ATIP-Avenir program from INSERM (R21042KS / RSE22002KSA) and the Emergence program from the University of Paris-Cité (RSFVJ21IDXB6_DANA).

## Author contributions
F.T., J.C., and A.B. conceived and performed this study. F.T., J.C., and A.B. designed DefenseFinder and most of the models. A.H. developed the package and the webserver. E.M. developed the data visualization. M.T. built the CRISPR-Cas models. F.T., J.C., and A.B. performed the analysis of the data and C.D. contributed to the analysis of the data. All the authors participated in the redaction of the manuscripts.

## Competing interests
The authors declare no competing interests.
