## [Peer Review File · Nature Communications]

Reviewers' Comments:

Reviewer #1:

Remarks to the Author:

In this work Tesson et al. provide systematic analysis of known anti-phage defense systems found in bacteria and archaea. They developed DefenseFinder, a tool that automatically detects and maps known anti-phage systems in microbial genomes. Since many anti-phage systems are co-localized in the same genomic position (i.e defense islands), this software could be used to aid in the discovery of new types or subtypes of systems.

This is a well written paper that presents a very useful new tool for the mapping of defense islands. The authors have addressed all of the comments/concerns from my previous review.

Reviewer #2:

Remarks to the Author:

To reiterate my review of the previous version of this manuscript, Tesson et al. provide a census of antiviral defense systems in prokaryotes. Any bioinformatics project can be improved technically (if, often, only marginally) by pouring more resources into the analysis (larger datasets, more sophisticated methods, etc.). In my opinion, the authors chose a reasonable point on the diminishing returns curve and the work is an acceptable compromise between the expense and accuracy.

The authors do not show any astounding novel results or generalizations; still both the current census and, more importantly, the DefenseFinder profile library would be quite valuable for the scientific community.

I am satisfied with the revised version of the manuscript.

Reviewer #3:

Remarks to the Author:

The revised manuscript by Tesson and Bernheim improved greatly compared to the previous version. Most of the issues have been solved but there are still points that need to be addressed.

The following changes are suggested:

- a) Depiction of the genomic locus of the defense system and its gene neighborhood would be useful for visual inspection and validation of the gene cluster.
- b) It would be helpful if the output would return both the DNA and the protein sequence
- c) Provide the latest version on Github page
- d) the tool does not appear to work with google chrome browser

Our previous concerns:

10) For SspABCD only one protein sequence per gene cluster was used with the reason that there was only one sequence known/provided. However, the study by Wang et al. (27 april, 2021) gives 2678 genomes plus locations where this system can be found. Thus, this system should be reanalyzed by the authors. Moreover, Wang et al. found 2678 instances of this system being present (8% of the analysed genomes), while the analysis of the authors found less than 5%. Please comment on this discrepancy.

In addition, SspABCDE is not present in any of the figures, but is mentioned in the paper as being one of the systems that is searched for. Can SspABCDE be included throughout?

Minor:

5) For the Stk2 system, it is annotated as just being Stk2, however, in literature, it shows that Stk1 and Stk2 are needed: "Thus, in the absence of stk1, we can still observe some protective effect of stk2, but ultimately stk1 is required for efficient immunity." - Depardieu et al., 2016

7) Reference 1 is used as 1 in the text instead of using the name of the group

a) For example here: "For example, for the 10 systems described in 1, where a detailed detection"

8) The names are shortened, but it would be better to have them written out. For example, IetAS is annotated as Gao_Iet. It is confusing when searching the systems in the figures.

New concerns:

In Supplementary Figure 1 (Hit score repartition for HMM of a single-gene systems).

Most single-gene thresholds seem to be very confident and clean, however, those of RT_Type IC, Drs1, Shedu, Avs4A, Avs2A, and AbiH, seem somewhat arbitrary, why were these thresholds chosen? For example, RT_12 could have also been put at 400.

CBASS type I predictions overlap with Pycsar predictions from othe defense system finder tools such as Padloc. Please comment on this in the discussion section.

We thank reviewers 1 and 2 for the comments about our current version of the manuscript. We provide below answers to the remaining questions of reviewer 3.

Reviewer #3 (Remarks to the Author)

The revised manuscript by Tesson and Bernheim improved greatly compared to the previous version. Most of the issues have been solved but there are still points that need to be addressed.

The following changes are suggested:

Depiction of the genomic locus of the defense system and its gene neighborhood would be useful for visual inspection and validation of the gene cluster.

We provide a depiction of the genomic locus on the webserver. The genomic locus depiction is only available when the input is a nucleic fasta file (as the genomic positions can only be inferred when such an input is given). As the command line interface was designed to analyze many (thousands) of genomes at the same time, we reasoned that generating many jpeg files would be computationally costly while only a few of these might be analyzed by the users. In previous versions of MaccyFinder, such a feature existed and was removed in more recent version for that reason. We thus encourage users who want a data visualization to use the webservice for specific locus they are interested in.

It would be helpful if the output would return both the DNA and the protein sequence

Currently DefenseFinder does not annotate the genomes on its own. When an input in nucleic fasta is provided on the webserver DefenseFinder, the genomes are first run on Prodigal and then the detection is made by DefenseFinder. In both cases, sequences can then be retrieved by the users using the ID of the hits. However, we noted that remark and will try include the feature in next versions.

Provide the latest version on Github page

The github page contains the last version of the models (in the models directory) as well as the last version of the command line package.

<https://github.com/mdmparis/defense-finder-models>.

<https://github.com/mdmparis/defense-finder>

the tool does not appear to work with google chrome browser

We tried on several computers (mac, windows) with google chrome and it was working. If the reviewer can indicate which version of Chrome is used (and raise an issue on this on github) it could help us try to narrow down on the issue and reproduce the bug.

Our previous concerns:

For SspABCD only one protein sequence per gene cluster was used with the reason that there was only one sequence known/provided. However, the study by Wang et al. (27 april, 2021) gives 2678 genomes

plus locations where this system can be found. Thus, this system should be reanalyzed by the authors. Moreover, Wang et al. found 2678 instances of this system being present (8% of the analysed genomes), while the analysis of the authors found less than 5%. Please comment on this discrepancy.

In addition, SspABCDE is not present in any of the figures, but is mentioned in the paper as being one of the systems that is searched for. Can SspABCDE be included throughout?

In the first version of DefenseFinder we misplaced SspBCDE as a subsystem of DND. Thanks to the reviewer's comment, we corrected this by creating a system named SspBCDE by adding SspBCDEFGH. We looked into the data available in (Wang et al., 2021). We downloaded every assembly used by Wang and colleagues and ran DefenseFinder Ssp system. Our results show a sensitivity of 97,9% on this database. However, running this model on RefSeq complete genome database, only 477 systems were found (2,1 % of the genomes). Even if the sensitivity seems to be good (97,9%), there is a big difference between the percentage of genomes with this system between Wang's and our results. This difference could be explained by the difference in the database species distribution. This is now explicitly mentioned in Supplementary Figure 3.

The SspBCDE system has been added in the figure in the previous version of the manuscript. It is found in Figure 1a, Figures 3a, 3d, 3e; Supplementary Figure 7.

Minor:

For the Stk2 system, it is annotated as just being Stk2, however, in literature, it shows that Stk1 and Stk2 are needed: "Thus, in the absence of stk1, we can still observe some protective effect of stk2, but ultimately stk1 is required for efficient immunity." - Depardieu et al., 2016

We agree that Stk2's ultimate action relies on Stk1. However, Stk1 is part of the core genome of the Staphylococci as mentioned by Depardieu and colleagues "*Another STK known as Stk1 (sometimes also named PknB or PrkC), present in all Staphylococci*". While Stk2 is part of the pangenome. Moreover, both genes do not form an operon and thus do not fit in the exact definition of system. We qualify Stk1 more as a necessary host factor (similarly for example to RecBCD for Retron Ec48) than as part of the defense system which is why we did not include it.

Reference 1 is used as 1 in the text instead of using the name of the group. For example here: "For example, for the 10 systems described in 1, where a detailed detection"

This was changed.

The names are shortened, but it would be better to have them written out. For example, IetAS is annotated as Gao_Iet. It is confusing when searching the systems in the figures.

We modified the figure 3 to use the name we have been using for the IetAS/Gao_Iet systems.

New concerns:

In Supplementary Figure 1 (Hit score repartition for HMM of a single-gene system).

Most single-gene thresholds seem to be very confident and clean, however, those of RT_Type IC, Drs1, Shed1, Avs4A, Avs2A, and AbiH, seem somewhat arbitrary, why were these thresholds chosen? For example, RT_12 could have also been put at 400.

Shed1 threshold has been set up using this method but also using ROC-Curve analysis using Doron's article data (See Supplementary figure 2).

For the different RT profiles, we choose a threshold that is already high (100 for RT_12 and 200 for RT_1_C1-2-3). In the end there are few proteins that hit above this threshold (y-axis is in log-scale). When we run DefenseFinder, all the RTs are run together (they are all very similar) and only the one with the best score is selected. So, after the selection using the GA cut threshold there is another selection by comparing with the other RT of DefenseFinder.

For AbiH we used the GA cut that was present in the pfam website.

For Drs1, Avs4A and Avs2A, the representation in the Supplementary figure 1 doesn't show exactly why we used this threshold. For these three HMM, we got profiles hit with a score higher than 3000. Those scores reduce the visualization of the GA choice. Here, we show for those three HMM, a zoom in the interesting portion of the graph. The scores were chosen using some zoom on the Supplementary figure 1.

Score repartition between 0 and 200 for Dsr1, Avs2A and Avs4A : a zoom on Supplementary figure 1. GA threshold were chosen using zoom inside the supplementary figure 1 for some systems.

CBASS type I predictions overlap with Pycsar predictions from othe defense system finder tools such as Padloc. Please comment on this in the discussion section.

Answer: Pycsar system and CBASS systems are closely related as described in (Tal et al., 2021). The main difference between CBASS and Pycsar is the nucleotide used by the cyclase. In CBASS, the cyclase is supposed to use A and G whereas in Pycsar it uses C and U. In DefenseFinder, we consider Pycsar as a subclass of CBASS as they have the same type of mechanism and the difference lies in the type of nucleotide used. The overlap of detection is linked to the proximity between the two cyclases.

Actions: We add a sentence in the discussion section: “For example, Pycsar systems were recently described, and are closely related to CBASS systems. They are already detected by DefenseFinder as a subtype of CBASS. The difference lies in the type of nucleotide used by the cyclase”